# IC-Health Project: Development of MOOCs to Promote Digital Health Literacy: First Results and Future Challenges

**Lilisbeth Perestelo-Perez** [1,2,3], **Alexandra Torres-Castaño** [4,*], **Carina González-González** [5],
**Yolanda Alvarez-Perez** [4], **Ana Toledo-Chavarri** [2,4], **Ana Wagner** [6,7], **Michelle Perello** [8],
**Stephan Van Der Broucke** [9], **Gonzalo Díaz-Meneses** [10], **Barbara Piccini** [11],
**Amado Rivero-Santana** [2,3,4], **Pedro Serrano-Aguilar** [1,2,3] **and on behalf of the IC**
**Project Consortium** [†]

[1] Evaluation Unit of the Canary Islands Health Service (SESCS), 38109 Tenerife, Spain;
   lilisbeth.peresteloperez@sescs.es (L.P.-P.); pseragu@gobiernodecanarias.org (P.S.-A.)
[2] Health Services Research on Chronic Patients Network (REDISSEC), 38109 Tenerife, Spain;
   anatoledochavarri@sescs.es (A.T.-C.); amado.riverosantana@sescs.es (A.R.-S.)
[3] Center for Biomedical Research of the Canary Islands (CIBICAN), 38320 Tenerife, Spain
[4] Canarian Foundation Health Research Institute of the Canary Islands, 38109 Tenerife, Spain;
   yolanda.alvarezperez@sescs.es
[5] ITED Research Group, Department of Computer Science and Engineering, University of La Laguna (ULL),
   38200 Tenerife, Spain; cjgonza@ull.edu.es
[6] Endocrinology and Nutrition Department, University Hospital Insular Materno-Infantil,
   35016 Gran Canaria, Spain; ana.wagner@ulpgc.es
[7] Research Institute of Biomedical and Health Sciences, University of Las Palmas de Gran Canaria,
   35001 Gran Canaria, Spain
[8] Consulta Europa Projects & Innovation, 35017 Gran Canaria, Spain; michelle.perello@consulta-europa.com
[9] Faculty of Psychology and Educational Sciences, Université Catholique de Louvain (UCLouvain),
   1348 Louvain-la-Neuve, Belgium; stephan.vandenbroucke@uclouvain.be
[10] Department of Economics and Business Management, University of Las Palmas de Gran Canaria,
   35001 Gran Canaria, Spain; gonzalo.diazmeneses@ulpgc.es
[11] Diabetology Unit, Meyer University Children's Hospital, 50139 Firenze, Italy; barbara.piccini@meyer.it
* Correspondence: atorrcas@sescs.es
† **IC-Health Project Partner Consortium:** Canary Islands Government—General Directorate for Public
   Health (Spain), Associazione Comitato Collaborazione Medica (Spain), University of La Laguna (Spain),
   Université Catholique de Louvain (Belgium), University of Udine (Italy), Ulster University (UK),
   Tallinn University (Estonia), University of Las Palmas de Gran Canaria (Spain), Consiglio Nazionale delle
   Ricerche (Italy), Scanbalt (Denmark), Meyer Children's Hospital (Italy), Consulta Europa Projects &
   Innovation (Spain), FUNKA NU AB (Sweden), European Health Management Association (Belgium).

**Abstract:** Digital health literacy (DHL) is the ability to search, understand and evaluate information from digital media and apply that knowledge to solve health problems. However, currently many citizens have not developed these skills, and this compromises not only the self-management of their health, but the possibility that health services are socially sustainable. The objective of this article was to present the objectives, activities and results of the IC-Health project whose objective was to develop a series of massive open online courses (MOOCs) to improve the DHL skills of European citizens. An exploratory report on DHL's current evidence was developed. Furthermore, a survey, focus groups and group interviews were conducted to determine DHL levels and the needs of population cohorts (children; adolescents; pregnant and lactating women; the elderly; and people affected by type 1 and type 2 diabetes). A participation strategy with end users was designed through a community of practice for the creation of MOOCs with the seven European countries that participated in the consortium. Thirty-five MOOCs were developed in eight different languages and a descriptive

and exploratory assessment of MOOCs was conducted with new participants. This first evaluation indicated that MOOCs can be an effective educational resource for DHL and a facilitator of shared decision-making processes. The process of co-creation of MOOCs, the components, the challenges and the opportunities identified in this European project could be useful for other developers of MOOCs who want to co-create interventions with beneficiaries in similar settings. Further longer-term actions are still needed to improve citizens' DHL.

**Keywords:** digital health literacy; MOOCs; open education; co-creation; eHealth; shared decision making

---

## 1. Introduction

The use of health content available on the Internet requires citizens to develop specific skills. Some are related to the use of digital technologies, that is, being able to handle a computer device, access information in a digital format, assess its credibility and relevance and be able to apply it to a specific context, which is known as digital literacy (DL) [1]. Similarly, health literacy (HL) has been defined as the social and cognitive competences that determine a person's level of motivation and ability to access, understand and use information in ways that promote and maintain good health [2].

The concept of digital health literacy (DHL) [3] emerges from a convergence of DL and HL as the ability of individuals to search for, find, understand and evaluate health information obtained from electronic sources and to apply the knowledge acquired to address or solve a health problem [4].

The development of these competences is crucial, because although the Internet has democratized access to health information, it is essential to educate and empower individuals to understand and use this information in an effective way, so that the Internet can be a support resource in health issues and not a source of inequalities in access to information. In this sense, the development of resources, tools, platforms that help develop these skills in citizens and even help prevent the emergence of some health conditions is still necessary. Some open education tools, such as massive online open courses (MOOCs) can be a useful tool in this regard [5].

The MOOCs, which are online courses within the field of open education that allow access to a large number of participants and greater interaction between them, have emerged in an era in which open education and long-life learning have been widely accepted. MOOCs use traditional learning methods (reading materials, videos, online exams), as well as interactive components such as user forums and discussions, which facilitate interactions between participants, facilitators and experts [6,7]. The evidence points to MOOCs being an effective tool to improve DHL skills [5].

The improvement of DHL and HL represents a challenge for both society and health services because people with low levels of HL may have difficulties in interpreting indications in written format, such as prescriptions or hospital discharge reports [1,2]. In addition, they have worse health outcomes compared to those with an adequate level of HL, such as: a higher risk of hospitalizations, a higher ratio of admissions to emergency departments, a lower adjusted quality of life score in patients with chronic diseases and less therapeutic adherence [8–10].

In this sense, health information resources, based on information and communication technologies (ICTs) have the potential to present quality information, which is complete, updated and in a format that is accessible to all citizens. However, the studies of the quality of health content websites, show that not all health content websites offer complete and up-to-date quality information. This situation is being made worse by the rapid dissemination of information through social networks [11]. As this proliferation of low-quality information represents a challenge for citizens, because in many cases it is misleading and leads to self-diagnosis and to the free search for possible treatments, it is considered essential to develop DHL skills that allow people to take a more active role in discerning what health information can be truthful and useful on a daily basis [12]. Despite the fact that organizations like Health on the Net (HON) have developed criteria for evaluating the reliability and trustworthiness of

online health information, not all health sites are evaluated. Furthermore, there is an ongoing debate on the appropriate classification tools to assess the reliability of health information online and how to create good quality health information. Consequently, the evidence highlights education and the development of the necessary skills in the users of this information as a priority [13].

From the perspective of health services, patients can turn the use of health information on the Internet into an experience to catalyze the empowerment of the patient, allowing them to assume a more proactive and positive role in their health. Therefore, the use of health information available on the Internet is already part of the health decision making of many patients [5].

The development of HL and DHL skills in citizens also acquires a crucial role from the perspective of social sustainability, because the promotion of their empowerment is a catalyst for the self-management of health, which will have an impact on healthier citizens and is therefore socially sustainable.

The term sustainability is increasingly linked to health issues, and this has to do with the fact that sustainability provides a framework within which it is possible to overcome all inequalities in terms of access to health services.

According to the WHO, sustainability implies the ability to meet the needs of the present without compromising the ability to meet future needs. This perspective implies the capacity of health services to use resources effectively to optimize the provision of health services, which includes guaranteeing the effective use and benefit of citizens [14].

Therefore, the improvement of the DHL is considered to be linked to the 2030 agenda for sustainable development, since most of the difficulties involve citizens being able to effectively use health information, especially that offered on the Internet, which also makes the improvement of such skills a priority issue from a sustainability perspective [15].

The European Commission formulated the eHealth action plan 2012–2020 in order to meet these challenges that citizens face with the massification of eHealth information and services (i.e., the use of information and communication technologies for health and well-being). In this plan, DHL is highlighted and promoted as the strategy to counteract the lack of knowledge among citizens about the benefits and challenges of eHealth [16].

Within the framework of the Horizon 2020 (H2020) program, the European Commission launched a call to promote the development of DHL initiatives in the member states using different resources based on ICTs, such as massive open online courses (MOOCs). In addition, the incorporation of more participatory and inclusive initiatives from the perspective of users such as user-centered design and co-creation, aims at incorporating the preferences and needs of users in the design and development of different types of educational resources [17].

In line with this call, the IC-Health project (Supplementary Information) was developed with the general objective of developing MOOCs to improve DHL levels in European citizens as an open education initiative for all. Besides, taking into account the European Digital Competence Framework, also known as DigComp (European Digital Competence Framework for citizens: https://ec.europa.eu/jrc/en/digcomp), these MOOCs have focused on the development of the four fundamental digital competencies of DHL: finding, understanding, and evaluating the knowledge acquired through electronic sources and applying that knowledge to solve health problems. The development of MOOCs was done using a co-creation methodology [11]. These courses have been developed in eight different languages for different population cohorts. The population cohort approach was in line with the idea of offering resources to improve DHL and favor life-long learning, highlighted as an objective within the 2030 agenda for sustainable development [15].

The objective of this article was to present the objectives, main activities and results of the IC-Health project, which aimed to develop MOOCs to improve the DHL skills of EU citizens.

## 2. Materials and Methods

For the development and achievement of the objectives of the IC-Health Project, a consortium of 14 partners was set up, representing 7 European countries (Spain, Italy, Belgium, the United Kingdom,

Sweden, Denmark and Estonia) and involving 7 universities and research centers, 1 government representative (project coordinator), 1 hospital, 1 non-governmental organization, 2 small and medium-sized companies and 2 European networks. The methodology followed in the IC-Health project was adopted by the different groups belonging to the consortium. In each country, adaptations were made when necessary according to the available resources.

The target populations of the IC-Health project included: (a) children (between 10 and 13 years old); (b) adolescents (between 14 and 18 years old); (c) pregnant and lactating women (PLW); (d) people over 60 years of age; and (e) people (both adults and children) with type 1 and type 2 diabetes. These populations were selected because they were expected to have profiles of use and the specific needs related to DHL. People who met the selection criteria were contacted through community centers, health centers, hospitals, reference health professionals and schools. In almost all cases, it was necessary to hold a meeting to present the project's objectives and request authorizations from the corresponding authorities. Those who were interested in participating were offered more detailed information about the project, and after agreeing to participate in the study, they signed an informed consent.

The actions carried out in the project were aimed on the one hand at analyzing population cohorts and the different scenarios around DHL among citizens. In order to do this, a literature review, an exploratory survey, focus groups and group interviews were developed. Subsequently, the practice communities in each country were formed with the different cohorts, in which the co-creation process of the MOOCs could be developed.

## 2.1. Literature Review

The first step in the IC-Health project consisted of a literature review resulting in a report on the key factors, barriers, facilitators and trends in the field of DHL. While due to the tight time schedule and the heterogeneous nature of the information in the form of scientific articles, reports, projects, it was not possible to conduct a full systematic review: the literature review involved a scoping review in which all project partners contributed by providing scientific articles and research reports, in addition to a search of online databases (PUBMED, Scopus and Google Scholar) [18]. This scoping review had a cross-cultural perspective, since the participating countries were asked to provide information from the local and national context, and thus collect contextualized information on the development of initiatives and resources available within the scope of the DHL. The steps followed in this scoping review were: (1) identifying the research question; (2) identifying the relevant studies; (3) study selection; (4) charting the data; (5) collating, summarizing and reporting the results [19].

## 2.2. Survey

An exploratory survey was conducted in the participating countries to explore the interest and needs of the participants; the adult version was modified for children (10–13 years) and adolescents (14–18 years). The results obtained in this survey complemented the findings made in the literature review and in the key report [20].

A total of 1642 responses were received, 862 from the adult cohorts: PLW: 249; people with diabetes: 223; the elderly: 390) and 780 responses from the child and adolescent cohorts (children: 384; adolescents: 396). Figure 1 shows the number of responses that were obtained in each country in each of the population cohorts.

For both adults and children, the survey included some sociodemographic information, in addition to the following sections:

1. Internet and computer use: the objective of this section was to discover the level of digital skills of the participants and the type of device they used the most. This section was developed within the framework of the project and consisted of 11 closed questions that were formulated based on the information available in the European framework of digital competence for citizens [21].
2. Health literacy: for this section, all the questions of the European Health Literacy Questionnaire HLS-EU-Q (short version: 16 and 6 items), developed in the frame of the European Health

Literacy Project 2009–2012, intended to measure the level of HL. This scale was fully respected since, despite the recent critical reviews, it is still acknowledged as the most stable and effective validated scale to measure HL. Furthermore, it has the advantage of already being used in the European context. Validated translations in some EU languages already exist [22].

3. Health and the Internet: in this case, the 8 items of the eHeaLS scale [23] were used (questions 10–17). This scale was complemented by additional ad hoc questions aimed at investigating the participants' frequency, motivations, feelings and habits in using the Internet for health-related purposes. On the whole, the section is composed of 21 closed-ended questions.

The results of the survey were exported and organized in a protected Excel file for later analysis. Statistical analysis included:

1. Data preprocessing analysis;
2. Analysis of data quality;
3. Checking for similarities and differences between key variables;
4. Descriptive indicators such as frequency, mean, median, standard deviation, and range;
5. Visualization analysis.

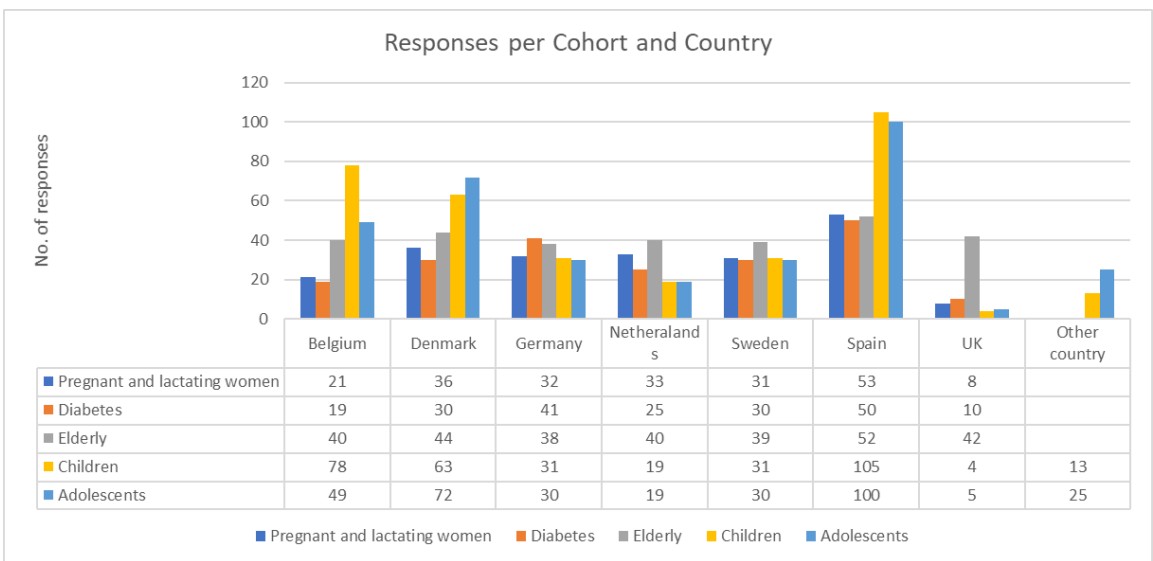

**Figure 1.** Number of responses per population cohort and country.

## 2.3. Focus Groups and Group Interviews

In order to complement the data collected in the survey, group interviews and focus groups were held in two of the countries of the consortium (Spain and Italy) to qualitatively explore the dimensions of DHL and to complement the information of the survey. Group interviews were carried out when only 4 to 7 people were recruited, and focus groups when 8 to 12 people were recruited.

A total of ten meetings were organized. In Spain, three group interviews and four focus groups were held, one for each of the following population cohorts: PLW, the elderly, adolescent, children, people with type 1 diabetes and people with type 2 diabetes. In Italy, one group interview and two focus groups were held in total. One in each of the following cohorts: PLW, adolescents with diabetes and the elderly [24].

A minimum of 4 and a maximum of 12 participants with varied socio-demographic profiles were invited; each group interview and focus group addressed a limited number of questions and lasted approximately two hours.

The following 4 topics were addressed:

- Use of the Internet in relation to health;
- Needs/expectations on the use of the Internet as a source of health information;
- Trust in the Internet as a source of health information;
- Other people's perception of Internet use in relation to health.

Given the exploratory nature of the survey, qualitative data gathered via the focus groups and group interviews were analyzed by means of a thematic analysis [25,26]. The following steps were followed in the focus groups and group interviews analysis: 1. in-depth analysis of the focus group/group interview audio-registration; 2. identification of the relevant debated themes/topics; 3. coding of each relevant theme/topic; 4. grouping of focus group/group interview information on a theme/topic basis; 5. critical analysis and interpretation of information gathered on each explored theme/topic; 6. incorporation of the focus group/group interview moderator's and assistant's observations; 7. synthesis of focus group/group interview results.

### 2.4. Co-Creation and Evaluation of MOOCs

The user perspective was incorporated through a co-creation methodology in all phases of MOOC design to ensure that their expectations, preferences and user needs regarding DHL were captured. The idea was to improve the quality of MOOCs by offering a participatory design process that took into account the users' perspective [27].

Communities of practice (CoPs) were the tool used to facilitate the co-creation process. CoPs are groups of people who share an interest in something they do and deepen their knowledge and experience on this subject through continuous interaction [28]. In the IC-Health project, the CoPs were made up of groups of people who belonged to the same population cohort (children, adolescents, PLW, older adults or people with diabetes) and were expected to have similar needs and interests. The objective of organizing a CoP for each of the cohorts in each country was to provide an environment (virtual or face-to-face) that would facilitate the co-creation process of the MOOCs through the exchange of ideas, experiences and the development of new knowledge from the contributions of all members. The members of the CoPs, and therefore those who co-created the MOOCs, were the representatives of the different population cohorts, health professionals (supporting groups of people with diabetes), educators (supporting groups of older people), teachers (supporting groups of children and adolescents) and facilitators' members of the project team.

Table 1 shows the results of the number of people involved in the co-creation process in each of the five cohorts in the participating countries of the project.

**Table 1.** People involved in the co-creation process by cohorts in each country.

|  | Spain | Italy | Belgium | UK | Sweden | Denmark/Germany/ Netherlands |
|---|---|---|---|---|---|---|
| Children | 56 | 96 | 17 | - | 20 | - |
| Adolescents | 31 | 44 | 17 | - | 20 | 25 |
| PLW | 56 | 16 | 5 | - | 20 | 9 |
| The elderly | 45 | 18 | 12 | 37 | - | 50 |
| Diabetes type 1 | 23 | 39 | 9 | - | 21 | 4 |
| Diabetes type 2 | 20 | - | 13 | - | 20 | |
| Total | 231 | 213 | 73 | 37 | 101 | 88 |
| Overall total | | | | 743 | | |

Once the CoPs were established in each participating country, the following steps were followed in the co-creation, which aimed to achieve a common vision of the participants regarding the content and format of the MOOCs.

1. Identification of a common, open, inspiring vision that offered a starting point for the creative process;

2.　Formulation of questions and the pros and cons of different scenarios to discover which was the best solution;

3.　Production of ideas and innovative alternatives regarding the different educational resources to be integrated into the MOOCs;

4.　Validation and evaluation of educational resources: the last step was the effective co-production of the materials, which meant the refinement of all the ideas and concepts generated in the previous step.

The evaluation of the MOOCs was done through questionnaires, both in paper and online format which focused on the following aspects: 1—the usability of the materials; 2—the effectiveness in improving the levels of DHL; and 3—the experience of the participants about the process of co-creation (Q15. Usefulness of co-creation approaches and methods used for the face-to-face activities, and Q16. Any relevant improvement or issue to report in relation to the co-creation process). Health professionals, educators, and graphic designers also participated in this evaluation.

Where possible, subjects were selected for their accessibility and proximity to the researcher, as this was the most appropriate and feasible method. Therefore, these data cannot be used to infer from the sample the general population in statistical terms.

The DHL was evaluated by the 8 items of the eHealth literacy scale (offline (before the use of the MOOC) and online (after the use of the MOOC)). The questions in this scale were related to the 4 competences of DHL. In order to evaluate the usability of the MOOCs, a question was incorporated at the end of each of the 35 MOOCs: *It is easy to navigate through this course, the information provided is presented in a clear manner.* With answer options ranging from "strongly agree" to "strongly disagree". Moreover, the face-to-face meetings were used to explore the qualitative aspects in greater depth, such as the users' navigation experience.

## 3. Results

### 3.1. Literature Review

The literature review identified the key aspects of interventions aimed at improving DHL levels:

### 3.1.1. The Universalization of Resources and Services through Digital Means Can Contribute to Inequality and Inequity in Access to These Services

DHL can help citizens to better manage their health, but the spread of digital tools to the health field can also increase inequalities in the access to this information. This is particularly true for those with limited DL, who will be at a disadvantage when searching for health information on electronic media. As reported in the Report of the eHealth Stakeholder Group on inequalities in access to health information and eHealth (produced by European Public Health Alliance, EPHA, 2014), a quarter of Europeans had never used the Internet at all. Interventions to improve DHL should therefore consider including extra content aimed at improving DL where necessary [29].

### 3.1.2. Low Health Literacy Linked to Poor Health Outcomes

The evidence consulted shows that low HL is related to poor health outcomes. In this respect, the evidence suggests that low levels of HL resulted in more hospitalizations, less participation in preventive examinations and vaccinations, low compliance with medication, the misinterpretation of health messages by health professionals, and higher mortality in the elderly. Another feature related to low HL is the difficulty in adequately understanding information. Inequalities may arise between those with higher levels of education and those with limited literacy, as indicated by Eurobarometer 404 [30].

### 3.1.3. Despite the Growth of the Internet as a Source of Health Information, Professionals Are the Most Valued Sources of Information

Recent surveys on DL show an increase in the search for and access to health information via the Internet in all age groups. Despite this increase in the use of the Internet for health issues, health professionals remain the most reliable source of information. This implies that if health professionals validate and incorporate digital resources, as tools to support the relationship with their patients, the result can be beneficial for both, since an increase in DHL can help in achieving better health outcomes in general [31].

### 3.1.4. The Readability, Clarity and Simplicity of the Content Found on the Internet Is a Relevant Issue When Searching for Health-Related Information

Linguistic, sociodemographic and cultural factors should be taken into account to avoid inaccurate interpretations of the information. These factors should be assessed in order to better adapt DHL interventions to specific populations and to tackle possible linguistic or cultural barriers [32].

### *3.2. Survey*

#### 3.2.1. Adults

The results in relation to the use of electronic devices, the Internet and DL show that most participants have access to the Internet and the most used devices are smartphones and computers, which are used on a daily basis. Emails and web browsing in general are the favorite activities. In addition, a correlation between the age and levels of Internet use is observed. PLW and people with type 1 diabetes access the Internet most, in contrast to older adults who use it the least. No significant differences were found between men and women in the levels of Internet access.

In general, participating countries have similar scores in DL, with only two exceptions: Denmark with the highest scores and Italy with the lowest. It should be mentioned that in Italy 90% of the survey participants were older adults.

As regards HL, all categories (finding, understanding, appraising, feeling of empowerment) show fairly similar results and the conclusion is that participants have a sufficient level of HL (above 60%) in general. The main difficulties mentioned were related to the management of mental health problems and the evaluation of the reliability of the information.

Regarding the DHL skills for health, the cohort of PLW, followed by the type 1 diabetes group, had the highest scores and the elderly and people with type 2 diabetes had the lowest scores. The type of information most frequently sought was about specific diseases, health problems or treatments and the reason is that they needed more information than that given by the clinician. When asked about the main perceived problem, participants from all the cohorts indicated that they were concerned about identifying truly reliable health information. In terms of cross-country comparison, Denmark and Germany had the highest scores in relation to DHL.

#### 3.2.2. Children and Adolescents

The devices most used by children are smartphones (76.3%) and tablets (71%), followed by the computer (64.6%). Almost all adolescents had a smartphone (94%), while 83.5% had a computer, followed by the tablet (63.6%). Furthermore, all the adolescents surveyed had one device or more, while in the case of children, some of them did not have any ICT device (4.4%).

In relation to DL, adolescents have better digital skills than children in general.

When interpreting the results regarding HL, the problems of quality of the measurement tool need to be considered. The sections related to health care do not apply to children. In the case of adolescents, most found it easy to obtain and understand health information, but some experienced difficulties in accessing and understanding information about preventing disease, promoting health, as well as in judging the reliability of information and applying it to support health-related decisions.

Regarding Health and Internet in children and adolescents, the most important conclusion is that a significant number of them have never looked for health information online because the most used source of health information is the clinician (83%) and their family and friends (66%), compared to the Internet (59%). In general, wellness and physical aspect information are the most searched topic. Finding information on the Internet about activities that promote mental health or about how to have healthier living habits or how to protect oneself or prevent diseases was described as "difficult". The highest values of Internet use for health were found in the groups of adolescents with diabetes in Italy, compared to the other groups of adolescents. Moreover, girls in general look for more information of this type than boys in all countries. The Danish and Swedish adolescents had the highest DHL skill levels.

### 3.3. Focus Groups and Group Interviews

The results presented are related to the focus groups/group interviews in Spain and Italy, as only these partners volunteered and managed to form a focus group or a group interview for each of the cohorts.

### 3.3.1. Children and Adolescents

Despite their familiarity with and use of devices (smartphones, tablets and computers) to surf the Internet, children and adolescents did not frequently use the Internet to search for health information. Some teenagers who had experience with searching for health information, pointed out that they did it for schoolwork or because they had a special health condition such as diabetes type 1. They mentioned that when they consult about health issues, they do so on topics that are not relevant, and before making a health decision, they discuss such information with someone from their family, friends, or health professionals. Although they liked the idea of accessing any information on the Internet quickly, they were concerned about the reliability and safety of the information. They were interested in a variety of topics, such as what to do about common diseases, healthy habits, and the health of their pets or pollution.

Children and adolescents mentioned that they preferred information in a short and simple format, adapted to their age and with resources such as gamification to make it more interesting.

### 3.3.2. Pregnant and Lactating Women

PLW were frequent users of the Internet to access health information on various topics: pregnancy, childbirth, early childhood, and nutrition. They indicated that this is more frequent in the first pregnancy. In their opinion, the Internet was a source of information, in a period full of doubts and when it is not always possible see a clinician. They feel confident about websites with well-structured information. They pointed out that the Internet can be of great help, especially when something goes wrong with their health professional. However, they believed that their clinician's opinion is essential about health information. PLW mentioned that official or accredited websites or those with many visits were the most reliable sources.

PLW pointed out that all topics related to pregnancy, childbirth and early childhood are valuable, with images and videos to make them more enjoyable, as well as placing importance on socio-cultural issues that can mean information is interpreted differently.

### 3.3.3. The Elderly

A differentiation can be made between two groups, in this cohort: those who use the Internet as a source of health information and those who do not use it because they do not trust the veracity of the information or because they lack digital skills. In general, most of the elderly stated that their clinician is the main source of health information. Among those who use the Internet, they do so mainly on sites recognized with official logos. Regarding their subjects of interest, they would like help tools to

be able to filter the health information and to support their decision about health. Among the most relevant topics they mentioned were cancer, cognitive exercises or information about diet.

The elderly would like simple and clear information with different support resources to make it easy to understand. The recommendations from their family members to help them searching for information on the Internet were also considered useful.

### 3.3.4. People with Diabetes

While the group of participants with diabetes included both people with type 1 and type 2 diabetes, a different user profile could be identified in each group. In the type 1 group, it was found that most used the Internet frequently to search for information that would help them to manage the disease, therefore these were people with high levels of DHL who felt empowered in the self-management of the disease. Their expectation of the MOOCs was to receive practical information and resources to manage the psychological aspects of the disease. In the type 2 group, the Internet was described as a secondary source of health information, with the main informational source being the health professional. Not being regular users of health information, their skills in DL, HL and DHL were minimal. This group highlighted a greater need to receive basic and quality information of the disease that would provide them with the possibility of correcting myths and false beliefs about the disease. In general, they mentioned that they trust accredited sites or sites with official logos.

Because of the different usage profiles in each group, it was necessary to develop two independent MOOCs that were sensitive to the characteristics and needs expressed by each group.

### 3.4. Co-Creation and Evaluation of MOOCs

Only general aspects of the MOOCs evaluation are presented in this section. As mentioned above, the detailed results of each of the cohorts will be analyzed in a series of future articles. Table 2 shows the list of MOOCs that have been co-created by the CoP members in each country: the groups of people from the cohorts and professionals as well as the project facilitators.

**Table 2.** List of the massive open online courses (MOOCs) developed in the IC-Health Project.

| Country | Number of MOOCs/Population Cohorts |
|---------|-----------------------------------|
| Belgium | 6 (children; adolescents; type 1 diabetes; type 2 diabetes; the elderly; PLW) |
| Germany | 1 (the elderly) |
| Denmark | 3 (adolescents; diabetes; PLW) |
| Netherlands | 1 (the elderly) |
| Italy | 9 (children; adolescents; type 1 diabetes; type 2 diabetes; the elderly; PLW) |
| Spain | 8 (children; adolescents; type 1 diabetes; type 2 diabetes; the elderly; PLW) |
| Sweden | 5 (children; adolescents; type 1 diabetes; type 2 diabetes; PLW) |
| UK | 2 (the elderly) |
| Total | 35 MOOCs |

### 3.4.1. Usability of Educational Resources

A high level of satisfaction (4/5) was observed regarding the usability of MOOCs. Most of the responses obtained related to the categories of agreement and strongly agree that *the MOOCs were easy to use*. However, the groups of children and adolescents reported a medium level of usability. Participants from all groups would recommend MOOCs. Figure 2 shows the results obtained in relation to usability in each of the cohorts.

Table 3 presents all the questions that were asked to users to determine their opinion about MOOCs (for details, please see in Appendix A, the Figures A1–A5 that illustrate the responses obtained in each of these questions by cohort.

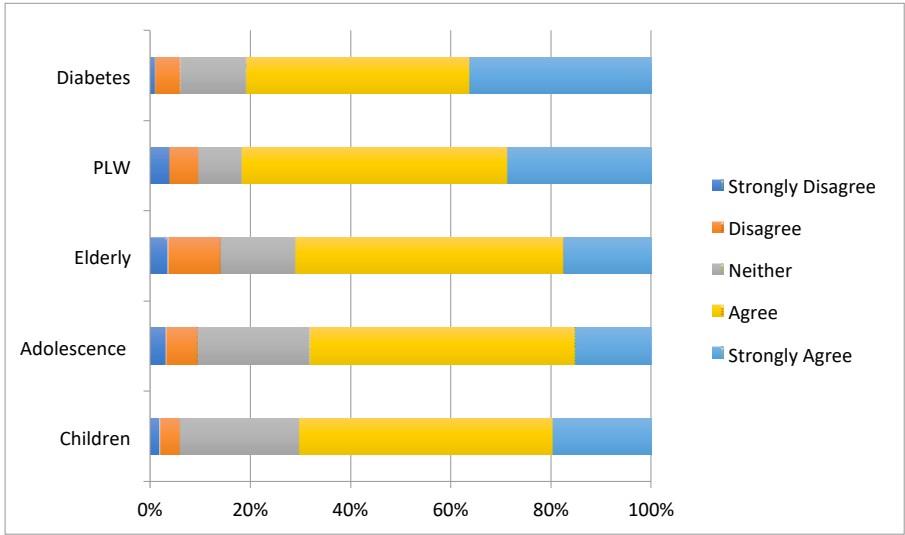

**Figure 2.** General feeling of usability in the cohorts.

**Table 3.** Questions on user opinion of MOOCs.

| | Opinion |
|---|---|
| O1 | This course has met my expectations |
| O2 | These objectives of the course were made clear. |
| O3 | The course content was consistent with the course objective. |
| O4 | Are the learning activities useful to gain a clear understanding of the course content? |
| O5 | How would you rate the quality/usefulness of the examples provided in the course? |
| O6 | The quizzes did appropriately test the material presented in the course. |
| O7 | How would you rate the overall design and aesthetics of the course content and materials? |
| O8 | I would recommend this course to other people. |

### 3.4.2. Level of Digital Health Literacy

The objective of the project was not to assess the changes of DHL (before and after the use of the MOOCs), since the DHL was assessed with independent samples. However, it should be mentioned that in all the cohorts evaluated, the scores obtained are slightly higher on the eHeaLS Scale in the measures taken after the use of the MOOCs, with respect to the measures taken before the use of the MOOCs. The results obtained in the open questions designed to objectively measure DHL are difficult to quantify categorically due to the diverse nature of the questions developed in each country. The different local teams presented an overall summary of the responses obtained, which indicated that 70–80% of the participants showed an excellent integration of DHL competencies by obtaining all the correct answers, after having completed the MOOCs.

### 3.4.3. Satisfaction with the Co-Creation Process

Fifty-six percent (56%) of the participants agreed that participating in the co-creation process made the final content of the MOOCs more relevant to their needs. Nineteen percent (19%) of the participants agreed completely and another 19% were not sure of the relevance of their participation in the co-creation process for the development of the courses. Half of the participants (54%) agreed that the co-creation process made them feel part of the project team. Twenty-three percent (23%) fully agreed with this statement, while 14% were unsure about this statement.

Regarding open questions (about the usefulness and possible aspects of improving the co-creation process), the participants in the adult groups emphasized that the co-creation process was important to them because it gave them the opportunity to share ideas and experiences with other participants,

and to play a key role in defining the structure and content of MOOCs. They also felt part of the project from the first face-to-face activity and appreciated the fact that their ideas were taken into consideration.

## 4. Discussion

The main goal of the IC-Health project was to co-design a series of MOOCs that could improve the DHL levels of EU citizens. Although the objective of the project has not been to generate changes in the participants regarding their DHL levels, the first MOOCs tests show promising early results in terms of DHL.

The next step, once 35 MOOCs were produced in eight languages, was to offer them as an open resource to be used by EU citizens and also to propose them for integration into other interventions to improve DHL. Doing this provided insight into what needs to be improved to ensure effective DHL interventions. As a result of the exploration with the survey, the focus groups and the group interviews, it became clear that one of the main concerns of users was trust in the health information available on the Internet, therefore the contents of the MOOCs focused on strategies for people to determine the quality of the websites autonomously.

The results obtained indicate that health professionals are the main source of health information for different population groups. However, the high levels of Internet use observed by PLW and people with type 1 diabetes suggest that the Internet may be another source of health information available to EU citizens, hence the importance of acquiring the skills to use this effectively. The results obtained indicate that MOOCs can be an effective educational resource for DHL.

Another relevant aspect that emerges has to do with the different user profiles and the different needs that they may have depending on whether they are a child, adolescent or adult or if they suffer from a particular health condition. This life-long learning approach implies the need for educational online resources, such as in the case of MOOCs, to be sensitive to the particularities of the users, both as regards aspects of usability and in concrete aspects of content.

Even though in terms of usability, the results show that the course was easy to use in practically all the cohorts, online educational interventions always run the risk of users dropping out. The option of nano MOOCs or short versions of MOOCs, such as those developed at IC-Health, have made it possible to avoid the problem of the dropouts. During the co-creation process of the materials, a combination of face-to-face and online meetings was created, which allowed for a greater connection between the participants and the COP, which in turn also served to avoid dropouts. Other aspects mentioned in the literature were included, such as taking into account the preferences of the person who was learning and the notion that the content responds to their objectives and motivations and is therefore relevant [6].

Furthermore, the results obtained from the evaluation of the process of co-creation of MOOCs by the participants indicate that inclusive and participatory methodologies generate a greater identification with the contents and a better evaluation of their relevance. With regard to the development of health content, the results indicate the need to take into consideration usability and accessibility criteria from the early phases of the design to ensure that the final product meets the characteristics, needs and preferences of the target groups. This aspect is crucial, since aspects such as the readability of content and the poor usability of DHL resources are considered to be one of the main barriers to accessing the health information available on the Internet [33].

Internet use for health issues implies a series of risks that are increased by the potential vulnerability of some user profiles. There are also a number of factors that increase that vulnerability, such as a desperate desire for a cure for their health condition, dissatisfaction with health care, and poor DHL, with HL and the DHL in particular having the most impact in the citizens. Among the populations most vulnerable to the risks of low-quality health information on the Internet are the elderly, youth and adults with low educational levels. Evidence indicates that these people use criteria far beyond those recommended in the guidelines to assess the quality of the websites [11]. Accordingly, it is crucial to promote the development of competencies that allow them to evaluate health information with criteria

on the quality and veracity of that information and to manage that information effectively and to be able to make informed decisions about their health, allowing an optimization in the provision of health services that not only has benefits at the individual level but also allows progress in solving inequalities in access to health information, as mentioned in the 2030 agenda for sustainable development [15,34].

The evaluation of MOOCs at DHL levels is only descriptive and exploratory, since the objective of the project was not to evaluate their effectiveness in terms of producing changes in the participants' DHL. Nevertheless, the early results obtained were considered promising since higher scores on the eHeaLS Scale in DHL are observed in all cohorts after the use of the MOOCs. In this sense, what is most relevant according to the objectives proposed in the project is the potential applicability of MOOCs as an open education tool, aimed at empowering citizens to improve their DHL, which is related to increasing their well-being and making them more socially sustainable.

Since DHL is a concept based on competence development (seeking, understanding, evaluating and applying), longer-term interventions are required to provide better results and with medium- and long-term effects [35]. The time frame of a European project, in which the central axis was the coordination of actions in a consortium of seven countries, makes it difficult to produce, develop and evaluate longer term interventions that will generate more evident changes in citizens.

## 5. Limitations

Regarding the exploratory survey, it should be mentioned that the responses obtained were not proportional to the countries participating in the project, nor to the population cohorts. Some of the consortium members reported difficulties in recruiting people and in meeting the objectives of this activity within the planned time frame. Likewise, it was only possible to carry out focus groups and group interviews in Spain and Italy because it was not an activity included at the beginning in the project's agenda.

Another aspect to mention is that the scales used, both in the survey and in the exploratory evaluation of MOOCs, with children and adolescents were adaptations of the adult scales, since it was not possible to find scales specifically designed for these populations.

Furthermore, as mentioned above, the objective of the project was not aimed at generating changes in the DHL levels of the participants, however, randomized controlled studies are needed to evaluate the effectiveness of the MOOCs on the DHL of EU citizens.

## 6. Conclusions

This project provides a practical example of the process of co-creating MOOCs to promote DHL. The process, components, challenges and opportunities identified could be useful for other developers and researchers who wish to co-create interventions with beneficiaries in similar settings. The interventions designed to improve DHL are crucial in a context where health information and services are multiplying. Despite the controls by national and international organizations and tools to filter misleading, low-quality information, the key remains the user of that information. Therefore, the greatest efforts should be directed at empowering the users, in the improvement of their HL and DHL and thereby improving their self-care behaviors and thus contribute to making the provision of healthcare more socially sustainable.

**Supplementary Materials:** Supplementary Information about the IC-Health project: https://ichealth.eu/.

**Author Contributions:** Conceptualization, L.P.-P., A.T.-C. (Alexandra Torres-Castaño) and C.G.-G.; methodology, L.P.-P., A.T.-C. (Alexandra Torres-Castaño) and M.P.; quantitative analysis, L.P.-P., A.T.-C. (Alexandra Torres-Castaño), M.P. and A.R.-S.; project management and qualitative analysis, A.T.-C. (Alexandra Torres-Castaño), M.P. and A.T.-C. (Ana Toledo-Chavarri); writing—original draft preparation, A.T.-C. (Alexandra Torres-Castaño), L.P.-P. and C.G.-G.; writing—review and editing, A.T.-C. (Alexandra Torres-Castaño), L.P.-P., C.G.-G., A.T.-C. (Ana Toledo-Chavarri), A.W., B.P., S.V.D.B., Y.A.-P., G.D.-M., M.P. and P.S.-A.; funding acquisition, L.P.-P., M.P., and the IC-Health consortium. All authors have read and agreed to the published version of the manuscript.

**Funding:** The IC-Health project has received funding from the European Commission's Horizon 2020 research and innovation program under grant agreement No 727474. The results and opinions presented here reflect

only the opinion of the authors and the Commission is not responsible for the use that can be made from the information it contains.

**Acknowledgments:** The authors wish to acknowledge and are grateful for the contribution of all the participants in this project for their effort and dedication.

**Conflicts of Interest:** The authors declare no conflict of interest.

## Appendix A

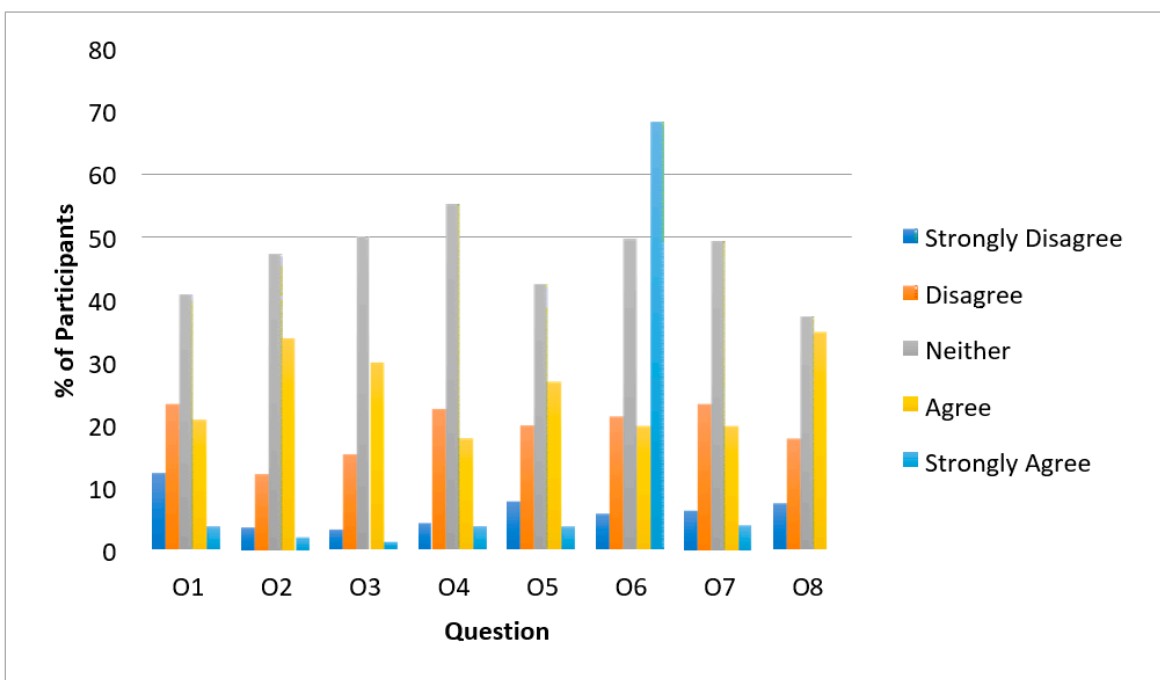

**Figure A1.** Children's responses in relation to their opinion regarding MOOCs.

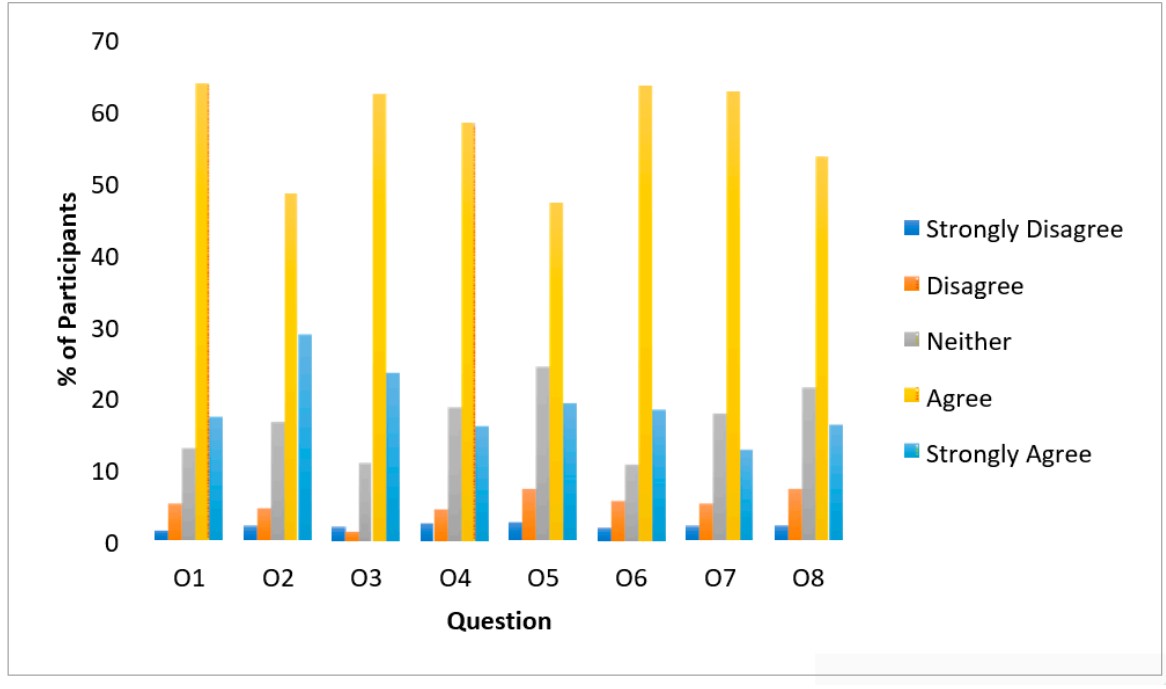

**Figure A2.** Adolescents' responses in relation to their opinion regarding MOOCs.

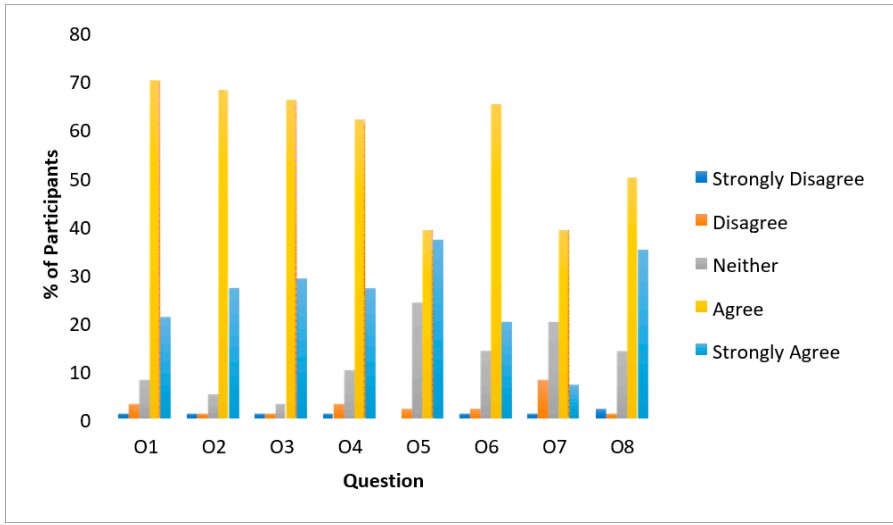

**Figure A3.** PLW's responses in relation to their opinion regarding MOOCs.

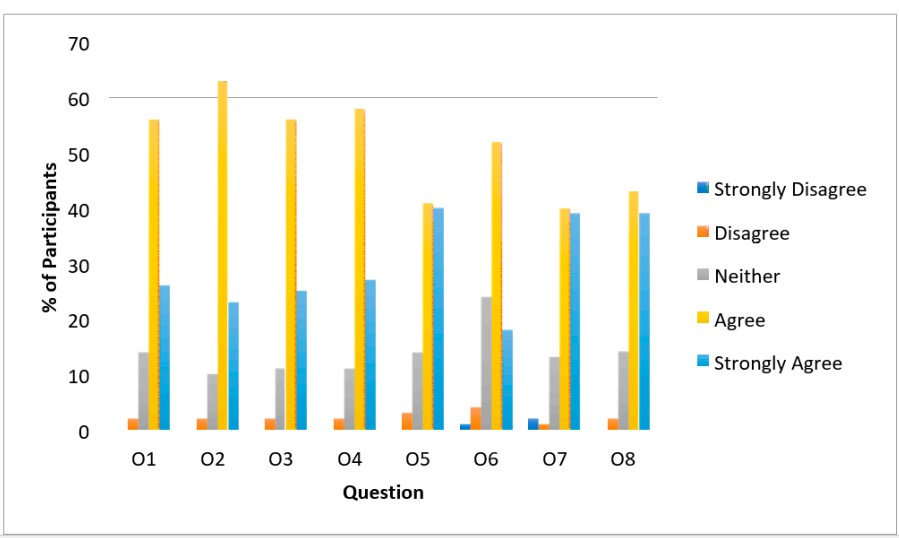

**Figure A4.** People with diabetes type 1 and 2's responses in relation to their opinion regarding MOOCs.

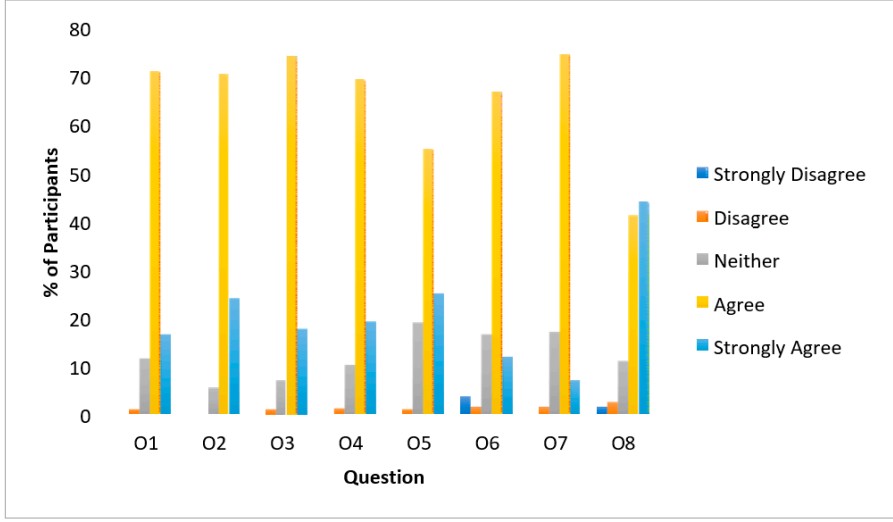

**Figure A5.** The elderly's responses in relation to their opinion regarding MOOCs.

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
