# Peer review of "IC-Health Project: Development of MOOCs to Promote Digital Health Literacy: First Results and Future Challenges"

_sustainability, doi:10.3390/su12166642_

Round 1
Reviewer 1 Report
The paper seems to be the first one from the intended series of papers illustrating the process and results of the IC_Health project. However, it seems that the authors have struggled with the material and encountered difficulties in terms of how to present the general overview of the project and how/how much to depict the results of the project. My suggestions for improvement and questions will be stated in hronological order in line with the presentation of the paper:
1) inaccurate title of paper - the paper does not cover the whole Europe, just some selected countries, besides, it is too general - the paper is more about co-creating MOOCs which are based on needs and skills of users. The paper does not tell us how much and in what way the DHL in general (not just in relation to MOOCS) has really changed in different countries (at least the main parts) of Europe. The title needs to be restated in order to match with the goals and essence of the paper.
2) p.3. 2.1.1. Literature review: this sentence is unclear:
While due to the tight time Schedule it was not possible to conduct a full systematic review, the literature review involved a scoping review in which all project partners contributed by providing scientific articles and research
reports, Please, explain, how did you really do the scoping review in a cross-cultural way.
3) p.4. 2.2. Exploratory survey. Please, add at least basic information about HLS-EU-Q16 questionnaire and eHeaLS scale.
4) p.4. - minimum of 4 participants does not sound appropriate for the focus group - explain the vindications for this choice.
5) p. 5. please, use the proper references to the methodological suggestions and type of qualitative data analysis you used for the transcripts of focus group discussions.
6) p.5. Did you involve in the CoPs any participants of survey or focus groups? If yes, mention it in the text of paper.
7) p. 5. All in all, who really created those 35 MOOCs?
8) p. 8. Regarding Health and the Internet in children and adolescents, the most important conclusion is that a significant number of them have never searched for online health information because the most used source of information are the clinician and their family and friends. Vague beginning pf sentence. How much exactly is this significant number?
9) p.8. You should speak about the gender differences also in case of the other samples.
10) p.9. Table 2 shows the list of MOOCs that were developed during the co-creation process in each country (See the list of tables in the appendix section). Developed by whom?
11) It seems that you have inserted just formal tables and figures in the paper (more to illustrate the methodological context), leaving our the most important information - results reflecting the real changes of participants' DHL.
12) p.10. Most of the 360 responses obtained related to the categories of agreement and strongly agree that the MOOCs were 361 easy to use. Please, quantify the results.
13) p.10. Participants from all groups would recommend MOOCs. Please, clarify - what participants (how many), from what groups, what about MOOCs - MOOCs in general, or specifically co-created MOOCs?
14) p.10. In all the evaluated cohorts there was an increase in the scores obtained on the eHEAL literacy Scale in the measurements taken when starting the MOOCs with respect to the measurements made
at the end of the MOOCs. Please, be more specific about cohorts and the mentioned changes do not sound logical - should be the opposite.
15) p.10. The post-evaluation provided a measure of the changes in participants' abilities to find, understand, apply, and evaluate health-related information via the Internet. The results obtained in the open questions designed to objectively measure DHL are difficult to quantify categorically due to the diverse nature of the questions developed in each country. The different local teams presented an overall summary of the responses obtained, which indicated that 70-80% of the participants showed excellent integration of DHL competencies by obtaining all the correct answers.
It is unclear - changes due to what? Participation in co-creation of MOOCs? Using newly created MOOCs or something else?
16) p.10 Table 3. Participants involved in the evaluation of the MOOCs
Cohort. Improper capture for Table 3. It would be better : The distribution of participants....Also, it is unclear - what is meant by the pre-assessment (of the MOOCs) and post-assessment (of the MOOCs)?
17) p.11. According to the evaluation of the impact of MOOCs on DHL levels, increases are observed in all cohorts. Unfortunately, you did not show these results properly in this paper. Improving your paper, please,, try to at least to some extent to show the quantitative prove of the impact of MOOC on DHL, using the results of statistical analysis.
18) p.12. Limitations. It seems that there were much more limitations than those mentioned here. Try to name them all.
19) p.12. Conclusions. The conclusions appear to be quite irrelevant to the project reflected in the paper. Please, rewrite them completely.
Besides these suggestions I would like to suggest the authors to find a way to embedd the paper in the context of sustainability. It would be just appropriate to add some connection between the DHL and social sustainability at the beginning and at the end of paper (at least).
Author Response
Thanks for all the comments that have allowed us to improve our article. Below I attach a document with the detailed responses to each comment.
Regards,

Reviewer 2 Report
The article undoubtedly presents important advances on health education and literacy in open education contexts such as MOOCs, platforms that have largely revolutionized lifelong learning. In this sense, the research presented is important, original, and will be a reference in the field of education and health.
Below are some recommendations to improve the quality of the manuscript:
- The abstract should be structured in a single paragraph and include the practical implications of the research for the future of health literacy and education.
- The article lacks a literature review, not specifically on DHL, but on MOOCs. It is important to understand that not all those who will read the manuscript are educators, so you should explain in detail and with original articles, the importance of MOOCs, its current usefulness. I recommend the following articles, which should be cited:
- Romero-Rodríguez, L.M., Ramírez-Montoya, M.S., & Aguaded, I. (2020). Determining Factors in MOOCs Completion Rates: Application Test in Energy Sustainability Courses. Sustainability, 12, 2893. http://dx.doi.org/10.3390/su12072893
- Ramírez-Montoya, M.S. (2019). Innovación en el diseño instruccional de cursos masivos abiertos (MOOC’s) para desarrollar competencias de emprendimiento en sustentabilidad energética. Education in the Knowledge Society (EKS), 20(5).
- Romero-Rodríguez, L.M., Ramírez-Montoya, M.S., & Valenzuela González, J.R. (2020). Correlation analysis between expectancy-value and achievement goals in MOOCs on energy sustainability: Profiles with higher engagement. Interactive Technology and Smart Education, Vol. ahead-of-print No. ahead-of-print. https://doi.org/10.1108/ITSE-01-2020-0017
- Cervi, L., Pérez-Tornero, J.M., Tejedor, S. (2020). The Challenge of Teaching Mobile Journalism through MOOCs: A Case Study. Sustainability 2020, 12(13), 5307; https://doi.org/10.3390/su12135307
- Figure 1 can be improved by using some DataViz software such as Tableau. This is a recommendation.
- The discussion and conclusions should be developed further in the light of the literature references set out above. The conclusions should also clearly state the practical implications of the research for the field of DHL.
Author Response

(The authors gave the same response as above.)
